# Estimation of Cadaveric Age by Ultrasonography

**DOI:** 10.3390/diagnostics10070499

**Published:** 2020-07-20

**Authors:** Hiroaki Ichioka, Daisuke Miyamori, Noboru Ishikawa, Risa Bandou, Nozomi Idota, Hiroki Kondou, Hiroshi Ikegaya

**Affiliations:** 1Department of Forensic Medicine, Graduate School of Medicine, Kyoto Prefectural University of Medicine, Kyoto 602-8566, Japan; ichioka@koto.kpu-m.ac.jp (H.I.); md199090@gmail.com (D.M.); ishikawanoboru@tdc.ac.jp (N.I.); risa1127@koto.kpu-m.ac.jp (R.B.); idotan@koto.kpu-m.ac.jp (N.I.); kondohrk@koto.kpu-m.ac.jp (H.K.); 2Department of General Internal Medicine, Hiroshima University Hospital, 1-2-3 Kasumi, Minami-ku, Hiroshima City, Hiroshima 734-8553, Japan; 3Department of Forensic Odontology and Anthropology, Tokyo Dental College, 2-9-18 Misaki-cho, Chiyoda-ku, Tokyo 101-0061, Japan

**Keywords:** bone mineral density, age, sex, Bone Area Ratio, deceased individuals, postmortem interval

## Abstract

(1) Background: While decreasing bone mineral density (BMD) with age in living people has been well documented, a correlation between age and bone mineral density in deceased people has only been reported in a few studies. A correlation between age and BMD in deceased people was investigated as an objective tool for age estimation of unidentified remains. (2) Methods: The Bone Area Ratio (BAR) was measured in 402 autopsy cases (143 females and 259 males over the age of 20). (3) Results: The correlation coefficient in the females was *r* = −0.5476, and the correlation coefficient in the males was *r* = −0.2166, indicating a stronger correlation in females than in males. A comparison of the BAR values in the deceased females for each age group with that in live females found no significant differences in the BAR values. BAR values in the deceased were similar to in live individuals, and this did not change with duration of the postmortem interval. (4) Conclusions: Measuring the BAR value based on bone mass using ultrasonic waves is rapid and easy, even for those lacking forensic training, and may be used to estimate the age of an individual and the likelihood of fracture due to trauma.

## 1. Introduction

Personal identification is necessary and important for unidentified remains found during either large-scale disasters or peacetime. When personal belongings and/or facial recognition are not available or conclusive, determination of both gender and approximate age from physical features is used for identification. Age estimation is determined from a wide variety of anthropological findings, such as adhesion of sutures between cranial bones, depth of humeral medullary cavities [1,2,3,4], dental attrition, and dental pulp cavity constriction [5]. Both physicians and dentists with forensic knowledge and experience are needed to make an age estimation. In-depth experience and broad professional knowledge are required to analyze the influence of occlusal tooth loss [6], and the presence of dental prostheses, dentures, and/or caries. Moreover, each forensic physician or dentist may reach different conclusions based on nonobjective evidence. Therefore, more precise objective methods, not necessarily requiring professional experience, are required for the determination of the age of deceased individuals.

While decreasing bone mineral density (BMD) with age in living individuals has been well documented [7], a correlation between age and BMD in deceased individuals has been reported in only a few studies [8,9,10]. Here, we propose that determination of a correlation between age and BMD in deceased individuals, as well as a review of the literature for the determination of age in living individuals, might enable use of the BMD as an objective estimation of age for unidentified remains. Furthermore, we investigated the use of BMD for the estimation of the age of unidentified remains at various postmortem intervals.

Ultrasonic waves and X-rays are used to measure BMD. Since ultrasonic waves do not require X-ray exposure and are convenient, this method is widely used in medical examinations, including for osteoporosis. The first clinical application of ultrasonic waves to bone mass measurement was in confirming a correlation between femoral cervical fractures and ultrasonic-wave-measured bone mass of the calcaneus, as reported by Langton et al. in 1984 [11]. Subsequently, a variety of studies utilizing ultrasonic waves for the calcaneus have shown efficacy in fracture evaluation [12,13,14,15,16]. Bone mass measurement using ultrasonic waves is used to estimate bone mass by measuring both the attenuation and velocity of the ultrasonic waves propagating in the bones. Devices using ultrasonic waves are popular for screening purposes, because they are objective, inexpensive, portable, and do not use radiation. Devices that use ultrasonic waves have many favorable features compared with the DXA method, which is a typical method for measuring bone density. Japan is the most frequent user of ultrasonic waves for evaluation of bone mass in the world [17]. This study examined the use of ultrasonic waves to determine a correlation between age and BMD in deceased and living individuals. The aim of this study was to investigate whether the correlation between age and BMD in deceased individuals can be used as an objective tool for age estimation of unidentified remains.

## 2. Materials and Methods

### 2.1. Subjects

This study involved 402 (143 females and 259 males over the age of 20) forensic autopsies with measured BMD, occurring between March 2011 and March 2017. The study was performed with the approval of the ethics committee at Kyoto Prefectural University of Medicine (approval number: ERB-C-1710-1,2020/05/25).

### 2.2. Bone Area Ratio (BAR)

The Bone Area Ratio (BAR) as the BMD of the right calcaneus was measured with a Benus α (Ishikawa Seisakusho LTD, Ishikawa, JAPAN) [18]. The Benus α, a bone mass measuring medical device approved in 1995, uses ultrasonic waves to determine the width and the ultrasonic wave transmission time of the calcaneus, and then calculates the BARs.

The BAR is specifically defined as follows: the length of bone area (a) with respect to the calcaneus width (b) is defined as the bone mass linear density (Eu). That is
Eu = Σa/b

The Benus α only measures the calcaneus horizontally. Since the trabecular bone anisotropy of the calcaneus is small, the value obtained by squaring the Eu obtained from the measurement results in one direction is defined as the BAR.
BAR = Eu^2^

The ratio of the trabecular bone to the cross-sectional area is shown.

### 2.3. Correlation between Age and BAR in Deceased Individuals

Spearman’s correlation coefficient was used to analyze age and measure BARs, for evaluation of a correlation between age and BAR.

### 2.4. Comparison with Data from Live Individuals

The data from deceased females was compared with that of 76 living females [19,20].

The Mann–Whitney U test was used to compare the BARs of the deceased individuals for each age group with the living individuals. The Mann–Whitney U test is a nonparametric test that tests differences in distributions of two unpaired data sets.

### 2.5. Analysis at Various Postmortem Intervals and Δ-BAR

A regression formula of age and BAR for the deceased individuals was first calculated from the samples (143 females and 259 males). The formula was used to calculate the BAR from the age of the identified samples with various postmortem intervals. A difference (Δ-BAR) in the BARs between that calculated from the formula and that measured by the Benus α was determined. A correlation between the postmortem intervals and the Δ-BAR in each sample was analyzed to determine the influence of postmortem intervals on the BARs. Spearman’s correlation coefficient was used to analyze the correlation between postmortem intervals and the Δ-BAR.

## 3. Results

### 3.1. Correlation between Age and the BAR in Deceased Individuals

The correlation coefficient between age and the BAR in the 402 total samples, including males and females, was *r* = −0.3572 (Figure 1a). The correlation coefficient in the 143 females was *r* = −0.5476, and the correlation coefficient in the 259 males was *r* = −0.2166, indicating a stronger correlation in females than in males (Figure 1b,c).

### 3.2. Comparison with Data from Living Individuals

The correlation coefficient between age and the BAR in living females [19,20] was *r* = −0.5182 (y = −0.1561x + 36.99), which was similar to that in the deceased females *r* = −0.5476 (y = −0.1506x + 38.71) (Figure 2).

A comparison of the BAR values in the deceased females for each age group with that in the living females found no significant differences in the BAR values (Figure 3).

### 3.3. Postmortem Intervals and Δ-BAR

Calculation of the regression formulae of age and BAR for the deceased individuals resulted in y = −0.00542x + 0.3681 for females and y = −0.00542x + 0.3681 for males, respectively. The correlation coefficients of postmortem intervals and the Δ-BAR, calculated from the regression formulae, were *r* = −0.01459 for females and r = 0.1027 for males, both indicating no correlation (Figure 4).

## 4. Discussion

This study identified BAR shows a stronger correlation in females than in males, which is why there is less influence of aging on BMD in males than that in females. This is due to more physical activity by men than women, who have a decline of BMD due to hormone changes during and after menopause.

The correlation coefficient between age and BAR in deceased females was close to that in living females. No difference was found in the BAR values between the deceased females for each age group and the living females. No correlation was found between the postmortem intervals and the Δ-BAR, suggesting no influence of postmortem intervals on BAR. Therefore, the BMD values in the deceased were similar to those in the living individuals, and this did not change with duration of the postmortem intervals.

Measuring the BMD value based on bone mass using ultrasonic waves is not accurate, according to our data. However, it is nondestructive, rapid, and easy, including for physicians and/or dentists lacking forensic training. It may support age estimation alongside other conventional methods. Having objective evaluable values for age estimation will allow easier and more rapid identification of age in deceased individuals, in not only large-scale disasters but also during peacetime.

Furthermore, measuring the BMD of samples in a forensic autopsy can be used to estimate the age of the individual and the likelihood of a fracture being due to trauma. 

Although bone mass measurements using ultrasonic waves to measure heel sizes, temperature, and edema have been reported [21,22], the autopsy samples in this study demonstrated a correlation between age and the BAR that was similar to the correlation found in living individuals. Hereafter, autopsy samples will be analyzed based also on heel sizes, temperature, and edema levels to determine the relationship of these parameters to bone mass measurements using ultrasonic waves.

In this study, the same result was obtained regardless of the operator for 10 measurements. Although the reliability is high, the accuracy of age estimation using ultrasonic waves is not. Therefore, this method should be used at the scene for rapid identification, and other conventional methods should be used in later examinations, including an autopsy. The number of autopsy samples that will be analyzed using ultrasonic waves in the future will further support and better define the estimation the age of an individual, and our basic data will be able to be utilized for research relating to living individuals.

## 5. Conclusions

The BMD values in deceased individuals were the same as in living individuals. The correlation coefficients of postmortem intervals and the Δ-BAR, calculated from the regression formulae, indicated no correlation. BMD values do not change with duration of postmortem intervals.

## Figures and Tables

**Figure 1 diagnostics-10-00499-f001:**
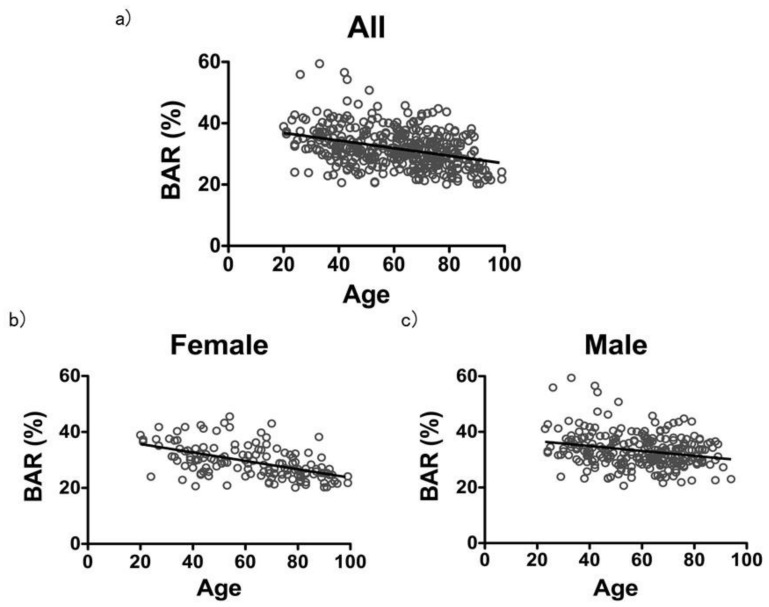
Correlation between age and Bone Area Ratio (BAR). (**a**) Correlation between age and BAR in all samples. The correlation coefficient between age and BAR in all samples, including males and females, was *r* = −0.3572. *n* = 402, Slope = −0.1246. (**b**,**c**) Correlation between age and BAR in females and males. The correlation coefficient in the females was *r* = −0.5476, and the correlation coefficient in the males was *r* = −0.2166. Female: *n* = 143, Slope = −0.1506. Male: *n* = 259, Slope = −0.08845.

**Figure 2 diagnostics-10-00499-f002:**
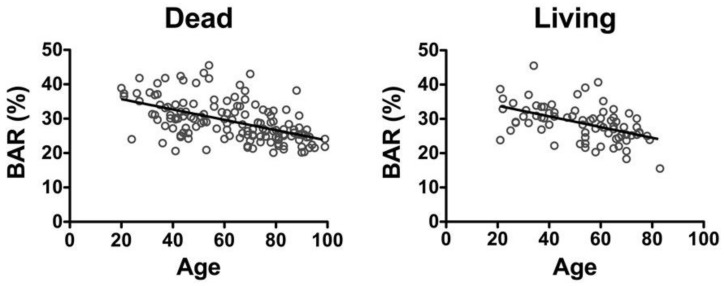
Correlation between age and BAR in deceased and living females. The correlation coefficient between age and the BAR in the deceased females was *r* = −0.5476 (Dead). The correlation coefficient between age and the BAR in the living females [19,20] was *r* = −0.5182 (Living). Dead: *n* = 143, y = −0.1506x + 38.71. Living: *n* = 76, y = −0.1561x + 36.99.

**Figure 3 diagnostics-10-00499-f003:**
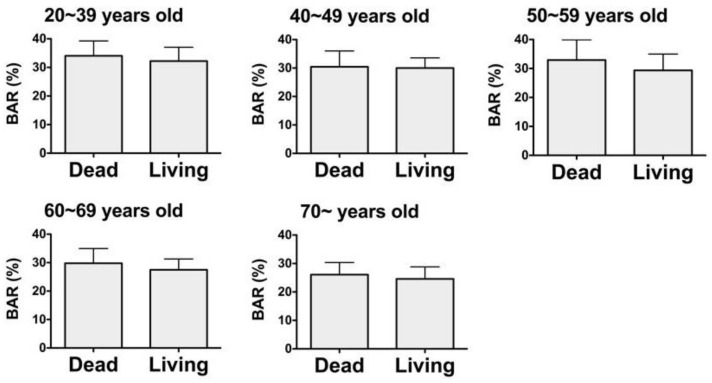
Comparison of the BAR values in deceased females for each age group with that in living females. No significant differences in the BAR values were found. Groups: 20~39 y.o. (years old), Dead; *n* = 15. 20–39 y.o., Living; *n* = 20. 40–49 y.o., Dead; *n* = 17. 40–49 y.o., Living; *n* = 8. 50–59 y.o., Dead; *n* = 11. 50–59 y.o., Living; *n* = 20. 60–69 y.o., Dead; *n* = 20. 60–69 y.o., Living; *n* = 19. 70- y.o., Dead; *n* = 52. 70- y.o., Living; *n* = 14. Mann–Whitney U test, mean ± S.D.

**Figure 4 diagnostics-10-00499-f004:**
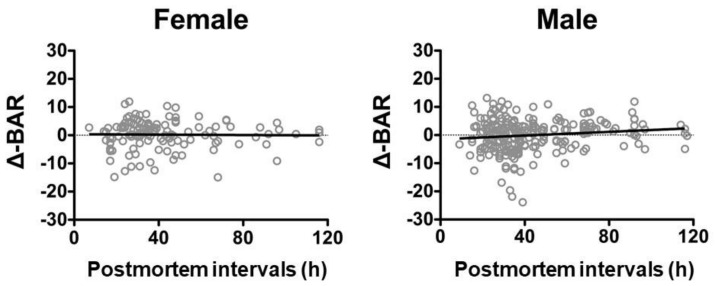
Correlation between postmortem intervals and Δ-BAR in the females and males. No correlations were found. Female: *n* = 127, Spearman *r* = −0.01459, y = −0.00542x + 0.3681. Male: *n* = 230, Spearman *r* = 0.1027, y = 0.03283x − 1.481.

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
