# Peer review of "Estimation of Cadaveric Age by Ultrasonography"

_diagnostics, 2020, doi:10.3390/diagnostics10070499_

Round 1
Reviewer 1 Report
Please see the attached PDF file for my comments.
Thanks

Author Response
Dear Reviewer #1:
We are grateful for the reviewer’s critical comments and useful suggestions that have helped improve our manuscript. As indicated in the following response, we have considered the comments and suggestions and have revised our manuscript accordingly.
Comment 1)
The article is very brief and it requires expansion in several parts:
- Introduction: Several important works are missing from the introduction. For instance, it would be beneficial to mention other methods in correlating BMD with age:
Paschall A, Ross AH. Biological sex variation in bone mineral density in the cranium and femur. Science & Justice. 2018 Jul 1;58(4):287-91
Botha D, Lynnerup N, Steyn M. Age estimation using bone mineral density in South Africans. Forensic science international. 2019 Apr 1;297:307-14.
Navega D, Coelho JD, Cunha E, Curate F. DXAGE: a new method for age at death estimation based on femoral bone mineral density and artificial neural networks. Journal of forensic sciences. 2018 Mar;63(2):497-503.
And more…
Answer
As the reviewer pointed out, we added more references (references 8-10) and modified sentences in “Abstract” and “Introduction”.
In addition, in response to the advice from the reviewer, we added the following sentence to the “Introduction”.
Devices using ultrasonic waves are popular for screening purposes because they are subjective, inexpensive, portable, and do not use radiation.
- Methods: It is far better that the paper be self-sufficient. That is why I strongly suggest that you include the definition of BAR, Mann-Whitney U, and your regression method (is it simple?) in the paper.
Answer
In response to the advice from the reviewer, we added the following sentences in “Materials and Methods”.
BAR is specifically defined as follows. The length of bone area (a) with respect to the calcaneus width (b) is defined as the bone mass linear density (Eu). That is,
Eu = Σa / b
The Benus α only measures the calcaneus horizontally. Since the trabecular bone anisotropy of the calcaneus is small, the value obtained by squaring the Eu obtained from the measurement results in one direction was defined as BAR.
BAR = Eu2
The ratio of trabecular bone to the cross-sectional area is shown.
The Mann-Whitney U test is a nonparametric test that tests differences in distributions for two unpaired data sets.
A regression formula of age and BAR for the deceased individuals was first calculated from the samples (143 females and 259 males).
- Discussion: Comparing your method with others
Answer
In response to the advice from the reviewer, we added the following sentences in “Discussion”.
In this study, the same result was obtained regardless of the operator throughout 10 measurements. Although the reliability is high, the accuracy of age estimation using ultrasonic waves is not. Therefore, this method should be used on the scene for rapid identification and other conventional methods should be used in later examinations, including autopsy.
- Conclusion: The correlation data is missing and it is in general too brief
Answer
In response to the advice from the reviewer, we added the following sentence in “Conclusion”.
The correlation coefficients of postmortem intervals and the Δ-BAR, calculated from the regression formulae, indicate no correlation.
Comment 2)
Please provide reasons for why have you selected calcaneus as the subject of your study and not another bone like femur.
Answer
Because stable reproducibility could be obtained even in a cadaver by conducting a study using a calcaneus that is popular for use in the living body. In response to the advice from the reviewer, we added the following sentence in “Discussion”.
in this study, the same result was obtained regardless of the operator throughout 10 measurements.
Also, because it is possible to compare with the data of the living body.
Comment 3)
The method you have implemented seems to be quite inaccurate. For example, I have tried to estimate an age based on three values of BMD on one of your plots and all of them led to a range between 35 to 90 years old, which is very rough.
This casts doubt on the accuracy of this method. But I understand that there are other inaccurate methods in this field too. Surely the method you suggest is easy, quick and inexpensive, but how does your work compare in terms of accuracy to other methods?
Answer
Measuring the BMD value based on bone mass using ultrasonic waves is rapid and simple even without special training. Devices using ultrasonic waves are portable. A linear model of the age-BMD correlation in females and males is practically useful, but as the reviewer pointed out, its accuracy for age estimation is not high. Therefore, this method should be used on the scene for rapid identification and other conventional methods should be used in later examinations, including autopsy. The combination of several methods at different times can be used to estimate the age. In the future, the number of autopsy samples analyzed using ultrasonic waves will increase and help to better estimate the age of the individual.
We added the following sentences to the “Discussion”.
Although the reliability is high, the accuracy of age estimation using ultrasonic waves is not. Therefore, this method should be used on the scene for rapid identification and other conventional methods should be used in later examinations, including autopsy.
Comment 4)
In the abstract and introduction, you claim that a study on correlation between BMD and age at death has not been reported. How about this study?
Navega D, Coelho JD, Cunha E, Curate F. DXAGE: a new method for age at death estimation based on femoral bone mineral density and artificial neural networks. Journal of forensic sciences. 2018 Mar;63(2):497-503.
I think this warrants a more thorough literature review in your paper.
Answer
As the reviewer pointed out, we added this reference and modified sentences in “Abstract” and “Introduction”.
Comment 5)
I think the meaning of the following sentence is not clear (Line 142): “Ultrasonic waves provide a socially-significant impact for families, investigators, governments, etc.”
Answer
As the reviewer pointed out, we deleted this sentence.
Comment 6)
Using evidence or articles, could you please back-up the proposed idea in Discussion that BMD can be used to evaluate probability of fracture?
Answer
As the reviewer pointed out, we added more references (references 15 and 16).
Finally, as the reviewer pointed out, we will postpone English editing until we submit the revised version of this manuscript. After that, we will request English Editing Service of MDPI.

Reviewer 2 Report
General comment: The authors presented an interesting work concerning to the estimation of cadaveric age by ultrasonography.
The manuscript should be revised by an English native.
Title: It is adequate.
Abstract: It is adequate.
Introduction: The Introduction is adequate. The aim of the study should be clearly stated at the end of the Introduction.
Materials and methods: They are adequately described.
Results: How many females were included in each age group. Please specify.
Discussion: The Discussion should be improved.
Recommendation: The manuscript should be accepted for publication after a moderate revision.
Author Response
Reviewer #2:
We are grateful for the reviewer’s critical comments and useful suggestions that have helped improve our manuscript. As indicated in the following response, we have considered the comments and suggestions and have revised our manuscript accordingly.
Comment
General comment: The authors presented an interesting work concerning to the estimation of cadaveric age by ultrasonography.
The manuscript should be revised by an English native.
Title: It is adequate.
Abstract: It is adequate.
Introduction: The Introduction is adequate. The aim of the study should be clearly stated at the end of the Introduction.
Materials and methods: They are adequately described.
Results: How many females were included in each age group. Please specify.
Discussion: The Discussion should be improved.
Recommendation: The manuscript should be accepted for publication after a moderate revision.
Answer
- As another reviewer pointed out, we will postpone English editing until we submit the revised version of this manuscript. After that, we will request English Editing Service of MDPI.
- As the reviewer pointed out, we added the following sentence in “Introduction”. The aim of this study is that a correlation between age and BMD in deceased was investigated as an objective tool for age estimation of unidentified remains.
- We indicate the number of females in each age group in legend.
- We modified “Discussion”. In response to the advice from the reviewer, we added the following sentences in “Discussion”. In this study, the same result was obtained regardless of the operator throughout 10 measurements. Although the reliability is high, the accuracy of age estimation using ultrasonic waves is not. Therefore, this method should be used on the scene for rapid identification and other conventional methods should be used in later examinations, including autopsy.
Round 2
Reviewer 1 Report
Dear authors,
Thanks for addressing most of the comments in my previous review. As I care about your work and I would like it to be appreciated by those who want to read and use it, I still recommend comparing your method with others' (how is it better? in which areas is it better and in which ones is it worse?), since it can add considerable value to your study. I appreciate your acknowledging in the paper that the method you propose is of limited accuracy and should be only used for quick evaluation, if need be.
The sentences added to the manuscript need some structural and grammatical revisions. Besides, in Line 57 of the new manuscript, you list 'subjective' as a favorable feature of ultrasound; however, it is 'objectiveness' that is a desirable feature, not subjectiveness.
Best
Author Response
Dear Reviewer #1:
We are grateful for the reviewer’s critical comments that have helped improve our manuscript. As indicated in the following response, we have considered the comments and have revised our manuscript accordingly.
Comment)
As I care about your work and I would like it to be appreciated by those who want to read and use it, I still recommend comparing your method with others' (how is it better? in which areas is it better and in which ones is it worse?), since it can add considerable value to your study. I appreciate your acknowledging in the paper that the method you propose is of limited accuracy and should be only used for quick evaluation, if need be.
The sentences added to the manuscript need some structural and grammatical revisions. Besides, in Line 57 of the new manuscript, you list 'subjective' as a favorable feature of ultrasound; however, it is 'objectiveness' that is a desirable feature, not subjectiveness.
Answer
As the reviewer pointed out, we considered that this method should be used on the scene for rapid identification and other conventional methods should be used in later examinations, including autopsy. In response to the advice from the reviewer, we added the following sentence to the “Introduction in Line 58”.
Devices which use ultrasonic waves have many favorable features compared with the DXA method, which is a typical method for measuring bone density.
In Line 57 of the new manuscript, 'subjective' was miswritten. 'objective' is correct.
The manuscript has undergone English language editing by MDPI.
Reviewer 2 Report
The manuscript should be accepted for publication.
Author Response
Dear Reviewer #2:
We are grateful for the reviewer’s critical comments.
The manuscript has undergone English language editing by MDPI.